# A Compared Study of Eicosapentaenoic Acid and Docosahexaenoic Acid in Improving Seizure-Induced Cognitive Deficiency in a Pentylenetetrazol-Kindling Young Mice Model

**DOI:** 10.3390/md21090464

**Published:** 2023-08-24

**Authors:** Yueqi Yang, Xueyan Wang, Lu Chen, Shiben Wang, Jun Han, Zhengping Wang, Min Wen

**Affiliations:** 1Institute of Biopharmaceutical Research, Liaocheng University, Liaocheng 252059, China; qiqiyang77@163.com (Y.Y.); xueyanwang0704@163.com (X.W.); 17781028137@163.com (L.C.); junhanmail@163.com (J.H.); bioactiveschina@163.com (Z.W.); 2School of Pharmaceutical Sciences, Liaocheng University, Liaocheng 252059, China; wangshiben@lcu.edu.cn; 3Pet Nutrition Research and Development Center, Gambol Pet Group Co., Ltd., Liaocheng 252000, China

**Keywords:** epilepsy, cognitive deficiency, EPA, DHA, ferroptosis, young mice

## Abstract

Epilepsy is a chronic neurological disorder that is more prevalent in children, and recurrent unprovoked seizures can lead to cognitive impairment. Numerous studies have reported the benefits of docosahexaenoic acid (DHA) on neurodevelopment and cognitive ability, while comparatively less attention has been given to eicosapentaenoic acid (EPA). Additionally, little is known about the effects and mechanisms of DHA and EPA in relation to seizure-induced cognitive impairment in the young rodent model. Current research indicates that ferroptosis is involved in epilepsy and cognitive deficiency in children. Further investigation is warranted to determine whether EPA or DHA can mitigate seizure-induced cognitive deficits by inhibiting ferroptosis. Therefore, this study was conducted to compare the effects of DHA and EPA on seizure-induced cognitive deficiency and reveal the underlying mechanisms focused on ferroptosis in a pentylenetetrazol (PTZ)-kindling young mice model. Mice were fed a diet containing DHA-enriched ethyl esters or EPA-enriched ethyl esters for 21 days at the age of 3 weeks and treated with PTZ (35 mg/kg, i.p.) every other day 10 times. The findings indicated that both EPA and DHA exhibited ameliorative effects on seizure-induced cognitive impairment, with EPA demonstrating a superior efficacy. Further mechanism study revealed that supplementation of DHA and EPA significantly increased cerebral DHA and EPA levels, balanced neurotransmitters, and inhibited ferroptosis by modulating iron homeostasis and reducing lipid peroxide accumulation in the hippocampus through activating the Nrf2/Sirt3 signal pathway. Notably, EPA exhibited better an advantage in ameliorating iron dyshomeostasis compared to DHA, owing to its stronger upregulation of Sirt3. These results indicate that DHA and EPA can efficaciously alleviate seizure-induced cognitive deficiency by inhibiting ferroptosis in PTZ-kindled young mice.

## 1. Introduction

Epilepsy, a prevalent chronic neurological disorder, exhibits a higher incidence rate among pediatric populations compared to adults [1]. Of note, recurrent unprovoked seizures in early life are detrimental to neurodevelopment and lead to cognitive impairment [2,3]. Anti-epileptic drug (AED) treatment is the main therapeutic approach for managing childhood-onset epilepsy. However, drug-resistant childhood epilepsy accounts for 20–30% of all cases of epilepsy, and the use of AEDs has numerous adverse effects on neurodevelopment which can aggravate the cognitive deficiency further [4,5]. As such, it is imperative to develop novel therapeutic agents and strategies to prevent or alleviate the cognitive deficiency induced by seizures.

At present, the precise mechanisms related to cognitive deficiency caused by epilepsy remain mostly unknown, which poses a great challenge to the treatment of epilepsy. Ferroptosis is a newly defined form of regulated cell death characterized by the accumulation of intracellular iron ions, leading to the accumulation of lipid peroxide [6]. It has been widely reported that ferroptosis is implicated in various neurological disorders, including traumatic brain injury [7], stroke [8], Alzheimer’s disease [9], Parkinson’s disease [10], and Huntington’s disease [11]. Recently, ferroptosis has also been observed in children with epilepsy [12]. However, the role and mechanism of ferroptosis in seizure-induced cognitive deficiency in childhood epilepsy remain largely unknown. Identifying the mechanisms and the role of ferroptosis in seizure-induced cognitive deficiency in childhood epilepsy will provide a novel approach to preventing and improving this issue.

Omega-3 polyunsaturated fatty acids (n-3 PUFA) play a critical role in brain development and have been claimed to produce beneficial effects on neurological disorders in early life. Preclinical and clinical studies found that supplementation of n-3 PUFA, docosahexaenoic acid (DHA), and eicosapentaenoic acid (EPA) reduced seizure frequency in patients with epilepsy resistant to drugs [13]. DHA, the primary fatty acid component of neurons, has demonstrated a wide range of neuroprotective effects in numerous studies [14]. Compared to DHA, lower levels of EPA (only about 0.1% of total fatty acids) are present in the brain and therefore receive less attention regarding their role in brain function [15]. Surprisingly, it has been discovered that EPA is more efficacious than DHA in the treatment of various neuropsychiatric disorders [16,17]. Based on these findings, it is reasonable to question whether EPA exhibits superior efficacy to DHA in the treatment of childhood epilepsy. However, few studies have investigated the anticonvulsant effects of EPA and DHA on childhood epilepsy. In addition, the effects of EPA and DHA on ferroptosis in childhood epilepsy have not been explored.

In the current investigation, the impact of DHA and EPA alone on seizure and cognitive deficiency was examined in a PTZ-kindling young mice (3-weeks of age) model. Further investigation was conducted to focus on ferroptosis and elucidate its molecular mechanisms in a PTZ-kindling young mice model, providing novel evidence for the rational use of DHA and EPA in childhood epilepsy.

## 2. Results

### 2.1. Effects of EPA and DHA on PTZ-Induced Seizures in the Young Mice Model

In this experiment, continuous intraperitoneal administration of PTZ (35 mg/kg) to mice induces a model of persistent and severe epilepsy. PTZ treatment resulted in an increase in seizure score and frequency, along with a shortened latency to seizure onset compared with the Con group (Figure 1A–D). However, supplementation with either EPA or DHA significantly decreased seizure scores (from the seventh PTZ injection) and frequency, accompanied by an extended latency to seizure onset compared to the PTZ group. Notably, EPA demonstrated superior and earlier efficacy than DHA in reducing seizure scores at the seventh PTZ injection(Figure 1A,B) (*F* = 10.76, *p* = 0.0025). These results reflect the different efficacy of EPA and DHA against PTZ-induced seizure in the young mice model.

### 2.2. Effects of EPA and DHA on Spatial Learning and Memory Ability in a PTZ-Kindling Young Mice Model

The Morris Water Maze (MWM) was employed to evaluate the spatial learning and memory ability of PTZ-kindled young mice. As shown in Figure 2A, the escape latency of the PTZ group mice was significantly longer than that of the Con group mice during training periods (day 3, day 4). Next, the memory ability of mice was evaluated through a spatial probe test after finishing the navigation test. Compared to the Con group, mice in the PTZ group exhibited a lower number of platform crossings and less time spent in the target quadrant, resulting in more time being devoted to finding the previous platform (Figure 2B–D). The administration of DHA or EPA significantly improved these parameters compared to the PTZ group mice. During training periods, the escape latency was shortened by treatment with EPA and DHA. The noteworthy aspect is that EPA played a more prominent role than DHA on the third day (*F* = 57.5, *p* < 0.0001). In addition, the mice in the EPA group demonstrated significantly longer durations in the target quadrant compared to those in the DHA group, indicating their superior spatial memory capabilities. The findings suggest that EPA and DHA have a beneficial effect on spatial learning and memory in young mice with PTZ kindling, with EPA showing potential advantages in improving their abilities.

### 2.3. Effects of EPA and DHA on Neurotransmitter Disorders, as well as Cerebral Levels of DHA, EPA, and AA in PTZ-Kindled Young Mice

As shown in Figure 3, a significant increase in glutamate (Glu) was observed, while γ-aminobutyric acid (GABA) and GAB (A) receptor A1 (GABARA1) were downregulated in the PTZ group as compared to the Con group. Administration of EPA and DHA resulted in an increase in GABA and GABARA1 levels, as well as a decrease in Glu levels. Notably, EPA demonstrated a better upregulating effect on GABA-GABARA1 compared to DHA (Figure 3A–C) (*F* = 57.5, *p* < 0.0001). In addition, after supplementation with EPA and DHA in young mice, the brain DHA (*F* = 28.5, *p* = 0.0003) and EPA levels were significantly increased (*F* = 24.09, *p* < 0.0001), while brain arachidonic acid (AA) levels were significantly decreased (*F* = 27.72, *p* < 0.0001) compared with those of Con and PTZ groups. As expected, the EPA group exhibited higher levels of brain EPA compared to the DHA group.

### 2.4. Effects of EPA and DHA on Neuronal Damage in Young Seizure Mouse Model

The neuronal nuclear protein (NeuN) is an exclusive marker of postmitotic neurons. Changes in NeuN expression are associated with neuronal degeneration [18]. NeuN immunohistochemical staining revealed decreased NeuN expression in CA1, CA3 and DG regions of the hippocampus in the PTZ group compared with the Con group. This suggests that neuronal degeneration is occurring. The NeuN-related neurodegeneration induced by PTZ can be ameliorated through supplementation with DHA and EPA, in which EPA showed superiority (Figure 4A). Brain-derived neurotrophic factor (BDNF), is a biomarker of neuronal plasticity and plays a critical role in neuronal development and function [19]. The protein levels of BDNF obviously declined in the PTZ group compared with that of the Con group, while supplementation with EPA and DHA similarly increased the levels of BDNF (*F* = 23.61, *p* = 0.0003). Accordingly, significantly decreased synaptic plasticity-related protein (postsynaptic dense protein 95) PSD95 and synaptophysin (Syn) were shown in the PTZ group mice. Notably, the administration of EPA and DHA resulted in an up-regulation of PSD95 (*F =* 8.14, *p =* 0.0081) and Syn (*F* = 6.54, *p* = 0.0152) protein levels, with EPA exhibiting a superior effect.

### 2.5. Effects of EPA and DHA on Hippocampal Iron Metabolism in a PTZ-Kindling Young Mice Model

In the current findings, a significant increase in total iron content (Figure 5A) and iron deposition (DAB-staining, Figure 5B) was observed in the hippocampi of the PTZ group compared to that of the Con group. Further analysis revealed significant upregulation of iron regulatory proteins 1(IRP1), transferrin receptor 1 (TfR1), and divalent metal-ion transporter-1 (DMT1), as well as downregulation of ferritin heavy chain 1 (FTH1) and ferroportin 1 (FPN1) in the PTZ group compared to the Con group (Figure 5C–H). These results demonstrated that PTZ injection caused hippocampal iron dyshomeostasist. Administration of EPA or DHA was found to modulate the disorders in iron metabolism-related proteins, thereby effectively inhibiting iron overload. Differently, DHA confers an advantage in reducing TfR1 expression (*F* = 17.84, *p* = 0.0007), whereas EPA exhibits superiority in upregulating FPN1 (*F* = 23.08, *p* = 0.0028) and more effectively preventing excessive iron accumulation.

### 2.6. Effects of EPA and DHA on Hippocampal Lipids Peroxidation in a PTZ-Kindling Young Mice Model

Compared to the Con group, PTZ treatment significantly increased MDA levels in the hippocampi of young mice (Figure 6A), indicating an accumulation of lipid peroxidation. Next, the levels of GSH and the protein expression of GPX4, xCT, and FSP1, which play crucial roles in detoxifying lipid peroxidation were analyzed further. As shown in Figure 6B–F, a significant decrease in GSH, GPX4, xCT, and FSP1 was observed in PTZ-kindled mice compared with the Con group. Supplementation with EPA and DHA comparably improved the aforementioned phenomena, indicating either EPA or DHA alone can inhibit PTZ-induced lipid peroxidation.

### 2.7. Effects of EPA and DHA on Sirt3/Nrf2 Pathway in a PTZ-Kindling Young Mice Model

To further explore the mechanism of ferroptosis in a PTZ kindling young mice model, we analyzed the alterations of the Sirt3/Nrf2 pathway in the hippocampal. As depicted in Figure 7, the PTZ group showed a significant decrease in Sirt3, *p*-Nrf2/Nrf2 ratio, and SOD2 levels, accompanied by reduced activity of SOD compared to the Con group, indicating suppression of the Sirt3-Nrf2 pathway. Meanwhile, treatment with EPA or DHA significantly increased the protein expression of Sirt3, SOD2 and the ratio of *p*-Nrf2/Nrf2. Among them, EPA was superior in promoting Sirt3 (*F* = 25.25, *p* = 0.0002).

## 3. Discussion

In this study, we found that administration of EPA or DHA alone could reduce seizure severity and frequency and cognitive deficiency in the PTZ-kindling young mice model. Further analysis revealed that either EPA or DHA was able to inhibit PTZ-induced neurotransmitter imbalance, neuronal damage, and ferroptosis by activating the Sirt3/Nrf2 pathway. It is worth noting that EPA demonstrated an advantage in reducing seizure-induced cognitive impairment by inhibiting ferroptosis, with a stronger enhancement of Sirt3 observed in the PTZ-kindling young mice model.

As a part of this study, we first examined the effect that DHA or EPA alone on seizure frequency in a PTZ-kindling young mice model. In accordance with our previous results [20], both DHA and EPA reduced PTZ-induced seizures, while EPA had a faster onset of action than DHA. The exact mechanism underlying the convulsant activity of PTZ remains elusive, but it is widely believed to stem from an imbalance between excitatory (Glu) and inhibitory (GABA) neurotransmission [21]. In the present results, both DHA and EPA exhibited inhibitory effects on PTZ-induced imbalance of Glu and GABA. Recently, it was shown that n-3 PUFAs (EPA and DHA mixture) could improve the GABAergic synaptic efficacy of stressed rats when they were restrained [22]. Accordingly, our findings show that supplementing with either DHA or EPA enhances the GABA-GABARA1 axis, with EPA showing superiority. It is noteworthy that the brain DHA levels were comparably elevated following supplementation with both DHA and EPA, while higher brain EPA levels were observed in the EPA group compared to the DHA group. This suggests that the higher brain EPA may partly contribute to the priority of EPA on the GABA-GABARA1 axis. These findings indicate that EPA or DHA may possess antiepileptic properties by regulating neurotransmitters (Glu and GABA), but with varying degrees of effectiveness.

Research has shown that epilepsy increases the risk of cognitive disorder in children [23]. Similarly, in the present investigation, PTZ kindling led to a decline in cognitive function, as evidenced by impaired performance on the MWM test. Previous studies observed neuronal damage in the brains of patients with recurrent seizures [24]. Consistent with previous study [18], significant diminished staining of NeuN was observed in the hippocampus of PTZ group mice, indicating that the occurrence of neuronal loss. Furthermore, synaptic plasticity plays a crucial role in the molecular processes underlying learning and memory, which are closely associated with various cognitive disorders such as epilepsy-induced cognitive dysfunction [25]. The levels of proteins involved in synaptic plasticity alter during epileptogenesis [26]. Synaptic plasticity-related proteins, especially Syn and PSD95 proteins, are two typical signature proteins of synaptic remodeling that can directly or indirectly reflect changes in synaptic function and structure [27]. PSD95 has been employed as a marker to indicate synaptogenesis and synapse loss. It has been reported that the levels of PSD95 are downregulated during epileptogenesis, which may be associated with neuronal death or dendritic spine loss in the hippocampus [28]. BDNF participates in neuronal survival and development, synaptic plasticity and memory [29]. In addition, BDNF can up-regulate the expression of PSD95 and Syn to modulate synaptic plasticity and memory [30,31]. Consistent with previous studies [18,28], the PTZ group exhibited a significant decrease in the expression of BDNF, Syn, and PSD-95 proteins in the hippocampus. While, either EPA or DHA treatment revealed significant increases in BDNF, Syn and PSD95 protein expression compared to the PTZ group. These above results imply that the pharmacological mechanism of EPA or DHA in cognitive disorders of epilepsy may be connected to enhanced BDNF, PSD95, and Syn expression and improved synaptic plasticity in the hippocampus. The endocannabinoid (ECS) system plays a crucial role in the regulation of synaptic plasticity, which is indispensable for neuronal development, learning, and memory formation [32]. Previous research has demonstrated that EPA and DHA contribute to brain repair through modulation of the endocannabinoid signaling pathway [33]. Therefore, we speculate that the beneficial role of EPA and DHA in enhancing spatial learning and memory capabilities can be partially attributed to their regulation of the ECS system.

Iron, the most abundant trace metal in the brain, plays a critical role in both neurodevelopment and brain function. Iron deficiency can hinder infants’ neurocognitive development, while cellular iron overload may promote lipid peroxidation and induce ferroptosis [12]. Generally, intracellular iron homeostasis is strictly regulated by a group of proteins, including IRP1, DMT1, TfR1, FTH1 and FPN1 [34]. Intracellular iron deficiency activates IRP1, which increases the expression of iron uptake proteins TfR1 and DMT1 while decreasing levels of iron store protein FTH1 and sole iron export protein FPN1, ultimately leading to an increase in intracellular iron. Conversely, adequate levels of iron reduce IRP1, TfR1, and DMT1 expression while upregulating FTH1 and FPN1, effectively reducing excessive free iron. It has been reported that seizures can cause dysregulation in iron metabolism [35]. In the present investigation, PTZ treatment resulted in dysregulation of iron metabolism. This was supported by a significant increase in total iron, IRP1, TfR1 and DMT1 levels, as well as a decrease in FTH1 and FPN1 expression, which is consistent with our previous study [20]. However, both EPA and DHA alone treatment effectively prevented hippocampal iron dyshomeostasis by regulating these iron metabolism-related proteins in PTZ-kindled young mice. Interestingly, treatment with DHA resulted in a lower expression of TfR1 compared to EPA, while EPA exhibited a stronger upregulation of FPN1 and lower levels of iron. These findings suggest that either EPA or DHA could inhibit iron dysregulation in young mice induced by PTZ kindling but through different mechanisms, with EPA exhibiting greater efficacy than DHA.

Ferroptosis is driven by ion-dependent lipid peroxidation (LPO) accumulation. PTZ injections have been widely shown to increase LPO levels in the brains of animals’ [36,37]. GPX4 is central to the regulation of ferroptosis as it can detoxify LPO in a GSH-dependent reaction. Once GSH is deficient or GPX4 is suppressed, LPO accumulates [38]. Recent research showed that significant decrease in GSH levels, a reduction in the GPX4 activity, and an increase in LPO in the blood of children with epilepsy [12]. FSP1 functions as an oxidoreductase parallel to GPX4 in eliminating lipid peroxyl radicals and counteracting ferroptosis through coenzyme Q and NADPH [39]. Accordingly, our present study revealed significantly reduced GSH, GPX4, xCT, and FSP1 levels, accompanied by an increase in iron deposition and lipid peroxidation (MDA), indicating the occurence of ferroptosis in the hippocampus of the PTZ-kindling young mice model. Administration of either EPA or DHA alone comparably inhibited PTZ-induced lipid peroxidation by enhancing the xCT-GSH-GPX4 axis and FSP1 expression. In addition, membrane lipids containing PUFAs, particularly AA, are preferentially oxidized in ferroptosis. In fact, DHA is the main PUFA in membrane lipids in the brain [14]. Therefore, we infer that the suppression of LPO may be partially attributed to the reduced brain AA levels following EPA and DHA administration.

Nrf2 is a crucial transcription factor involved in the regulation of antioxidative gene expression. The activation of Nrf2 alleviates epilepsy severity and prevents spontaneous seizures [40]. Under conditions of oxidative stress, Nrf2 is activated and subsequently triggers a cascade of target genes responsible for cellular antioxidant response, including GPX4, xCT and SOD [41]. Meanwhile, Nrf2 activation increases cellular NADPH levels, which may facilitate the FSP1-mediated detoxification of LPO. Additionally, Nrf2 facilitates iron sequestration and attenuates cellular iron uptake [42,43]. Sirt3, a member of NAD+-dependent deacetylases, is localized to mitochondria where it functions in the deacetylation of numerous enzymes involved in response to oxidative stress, including SOD. Dysfunction of Sirt3 contributes to mitochondrial dysfunction in chronic epilepsy [44]. It is worth noting that Sirt3 plays a key role in regulating iron metabolism through mitochondrial ROS-IRP1 [45]. In addition, the overexpression of Sirt3 promotes the activation of Nrf2, which subsequently induces the expression of Sirt3, thereby augmenting its functionality [46]. Therefore, the Sirt3-Nrf2 pathway plays a crucial role in ferroptosis. Accordingly, a reduced Sirt3, *p*-Nrf2/Nrf2 ratio concomitant with decreased levels of downstream antioxidant enzyme SOD2 and diminished SOD activity were exhibited in PTZ kindling young mice. While, supplementation of EPA or DHA significantly reversed these phenomena. Cheng et al. [47] reported that Sirt3 preserves GABAergic interneurons and protects cerebral circuits against hyperexcitability in AD model mice. It is noteworthy that EPA demonstrated a superior effect in up-regulating Sirt3 compared to DHA, which may be contributed to its superiority in inhibiting iron overload and better efficacy in enhancing the GABA-GABARA1 axis. These above results indicate that both EPA and DHA inhibited PTZ-induced cognitive deficiency by activating the Sirt3-Nrf2 pathway in the PTZ kindling young mice model.

## 4. Materials and Methods

### 4.1. Animals and Study Design

The experimental protocol was approved by the Research and Ethics Committee of Liaocheng University’s Institute of Biopharmaceutical Research (Approval No. SWZY2020812). and followed the National Institutes of Health guidelines (NIH publications No. 80–23). The PTZ-induced mouse model is capable of replicating certain neuropathological features observed in human temporal lobe epilepsy and has been extensively utilized in animal studies [21]. A total of forty male ICR mice, aged 3 weeks, were randomly allocated into four groups, with each group consisting of ten mice. These groups included a control group (Con), a PTZ kindling group (PTZ), a PTZ kindling + EPA group (EPA), and a PTZ kindling + DHA group (DHA) (*n* = 10). Diets containing 1% EPA-enriched ethyl ester or DHA-enriched ethyl ester (70% EPA + DHA, *w/w*) were fed to the groups receiving either EPA or DHA, respectively (Figure 8). The dosage is consistent with the safe dosage mentioned in a previous clinical study, which is equivalent to 4g per day according to the conversion of human clinical dosage [48]. Appendix A shows the ingredient compositions and fatty acid profiles of experimental diets.

### 4.2. PTZ-Kindling Young Mice Model

The mice were kindled through repeated administration of PTZ (35 mg/kg via i.p. injection) [20] every other day for a total of eleven injections, with successful kindling defined as the occurrence of more than three consecutive stage 4 seizures. The behavioral alterations of mice were monitored for a duration of 30 min following the administration of PTZ. The assessment of behavioral seizures was conducted utilizing the 5-point Racine Score system. Specifically, Racine score I was indicative of facial clonus, score II denoted head nodding, score III represented unilateral forelimb clonus, score IV described rearing with bilateral forelimb clonus, and lastly, score V referred to rearing and falling accompanied by loss of postural control. The analysis of seizure stage, seizure onset latency (in seconds), and number of seizures was conducted in a blinded manner. The seizures were evaluated based on the established Racine scoring system [49].

### 4.3. The Morris Water Maze

The Morris Water Maze (MWM) test was conducted in accordance with previously published methods [9], with the water temperature maintained at approximately 25 °C in a large circular black pool for MWM testing. The pool is partitioned into four quadrants, with one quadrant featuring a concealed platform situated one centimeter beneath the water’s surface. The mice were trained for 4 days with 60 sec per trial to locate the hidden platform. The probe trial was conducted without the platform. The ANY-maze program (Stoelting Co., Wood Dale, IL, USA) was used to track and analyze the behavior tests.

### 4.4. Analysis of the DHA EPA and AA Levels of Hippocampi

Isolated hippocampi were immediately weighed on ice, followed by lipid extraction using previously established methods [50]. Gas chromatography (GC) was used to analyze the levels of DHA, EPA and AA in the hippocampi, expressed as a percentage of total fatty acids, as previously reported [40].

### 4.5. Analysis of Neurotransmitter

The extraction of neurotransmitters was performed according to a previous study [20]. We quantified the concentrations of two neurotransmitters, specifically γ-aminobutyric acid (GABA) and glutamate (Glu). Hippocampus was homogenized with ice cold methanol centrifuged at 12,000× *g* for 20 min at 4 °C. The methanol layers were collected, freeze-dried, and reconstituted with deionized water to a volume of 300 μL. Subsequently, 300 μL of chloroform isopropanol (100:30, *v/v*) was added followed by vortex mixing and centrifugation at 13,000× *g* for 5 min. The supernatant was analyzed by injecting a 5 μL aliquot into an Agilent 1200 Series LC system coupled to high-resolution mass spectrometry (AB Sciex, Darmstadt, Germany) for chromatographic separation on a SEPAX GP-C18 column (2.1 × 150 mm, 3 μm) at 25 °C. The mobile phase was composed of 90% A (0.1% formic acid and 20 mmol/L ammonium acetate) and 10% B (acetonitrile), with a flow rate of 0.25 mL/min for a duration of 6 min. Positive mode electrospray ionization was employed on the AB API 4000 mass spectrometer for multiple reaction monitoring. It was set at 450 °C, with 4.5 kV for the ion spray voltage. Multireaction monitoring mode was used for quantification, with mass analysis parameters listed in Appendix A.

### 4.6. Biochemical Analyses

To gain a comprehensive understanding of oxidative stress and cellular damage in the hippocampus, we employed kits procured from the Nanjing Jiancheng Bioengineering Institute located in Nanjing, China. These kits enabled us to accurately measure and analyze levels of total iron (A039-2-1), GSH (A006-2-1), MDA (A003-1), and SOD activity (A001-1).

### 4.7. Hematoxylin- and Eosin (H&E) Staining

A hematoxylin and eosin staining method was used for the histopathological examination of tissue sections embedded in paraffin (G1076-500ML, Servicebio, Wuhan, China). A bright-field microscope (Olympus, Tokyo, Japan) was used for the observation and recording of the sections.

### 4.8. Nissl Staining

The sections were obtained by cutting with a freezing microtome, stained with toluidine blue (G1036-100ML, Servicebio, Wuhan, China), and subsequently dehydrated in 30% sugar solution before being covered with 50% glycerin. Images were taken with a light microscope (Olympus, Tokyo, Japan).

### 4.9. Immunocytochemistry Assay

Immunohistochemical studies were performed following the previously established protocol [51]. Antibodies against MBP (1:1000; #78896; Cell Signaling Technology, Danvers, MA, USA) were applied to sections and left overnight at 4 °C. For the following 3 h at a temperature of 4 °C, they were treated with secondary antibodies that had been conjugated with horseradish peroxidase (Epizyme; Shanghai, China). The sections were examined under a light microscope (Olympus, Tokyo, Japan) and corresponding images were captured for further analysis.

### 4.10. Western Blot

To acquire the requisite proteins for our research, we utilize a modified RIPA buffer to lyse the tissue, as previously outlined in our study [9]. The primary antibodies of Sirt3(ab217319, 1:1000), TfR1 (ab214039, 1:1000), DMT1 (ab55735, 1:1000), FTH1 (ab183781, 1:1000), xCT (ab175186, 1:1000), and FSP1(ab197896, 1:1000) were purchased from Abcam (Cambridge, MA, USA), the primary antibodies of IRP1 (bs-9848R, 1:1000), FPN1 (bs-21360R, 1:1000), GPX4 (bs-3884R, 1:5000), Nrf2 (bs-1074R, 1:1000), *p*-Nrf2 (bs-23531R, 1:1000), and SOD2 (bs-1080R, 1:1000) were purchased from Bioss (Beijing, China), and the primary antibodies of *β*-actin (66009-1-Ig, 1:2000) were purchased from Proteintech (Chicago, IL, USA). In this study, the secondary antibodies used are HRP-conjugated Affinipure Goat Anti-Mouse IgG (H+L) (Cat No. SA00001-1, 1:5000) and HRP-conjugated Affinipure Goat Anti-Rabbit IgG (H+L) (Cat No. SA00001-2, 1:5000) were purchased from proteintech (Chicago, IL, USA). ECL Western blotting substrate was utilized for the development of blots, while the UVP Auto Chemi Image system (Tanon 4600SF, Shanghai, China) was employed to visualize luminescence.

### 4.11. Statistical Analysis

Data are presented as means + SEM. All analyses were adjusted for multiple testing using the Bonferroni and *p* < 0.05 was considered statistically significant.

## 5. Conclusions

In conclusion, our study has confirmed that the administration of EPA or DHA alone can improve seizures and cognitive deficiencies in the PTZ-kindling young mice model by inhibiting ferroptosis through the Sirt3-Nrf2 pathway. Notably, EPA demonstrated a greater efficacy than DHA. These findings suggest that targeting ferroptosis may prevent seizure-induced cognitive deficiency, and dietary interventions involving EPA or DHA could be a more effective approach.

## Figures and Tables

**Figure 1 marinedrugs-21-00464-f001:**
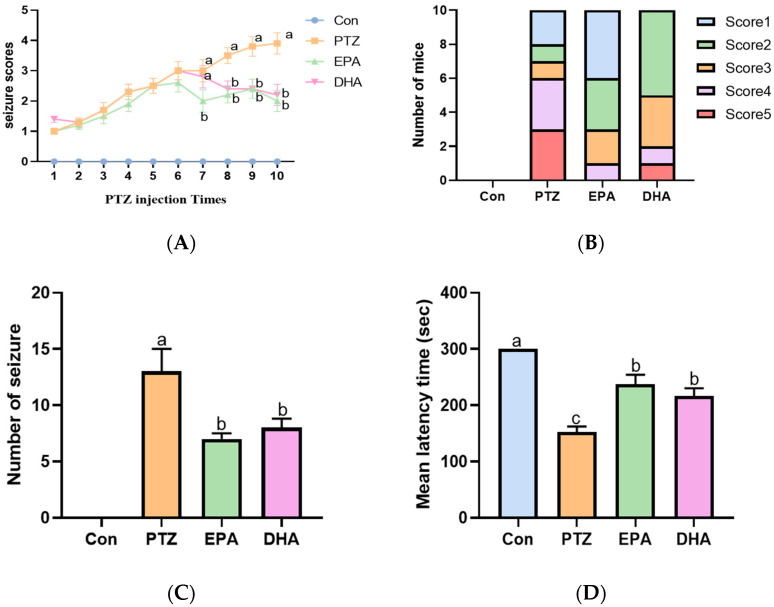
Effects of EPA and DHA on PTZ (35 mg/kg·bw, i.p.) induced seizure. (**A**) The seizure scores in mice after PTZ treatment every other day; (**B**) the seizure scores distribution among mice in different groups after the 7th PTZ treatment. (**C**) the number of seizures (Stage 4 or greater) in mice after PTZ treatment every other day; (**D**) the latency to major seizure of Stage 4 or greater in mice after PTZ treatment every other day. Data were expressed as mean + SEM (*n* = 10). The tested groups underwent repeated measures ANOVA. The Bonferroni post hoc test was employed to calculate *p*-values based on the Bonferroni-adjusted α. There are significant differences between different letters.

**Figure 2 marinedrugs-21-00464-f002:**
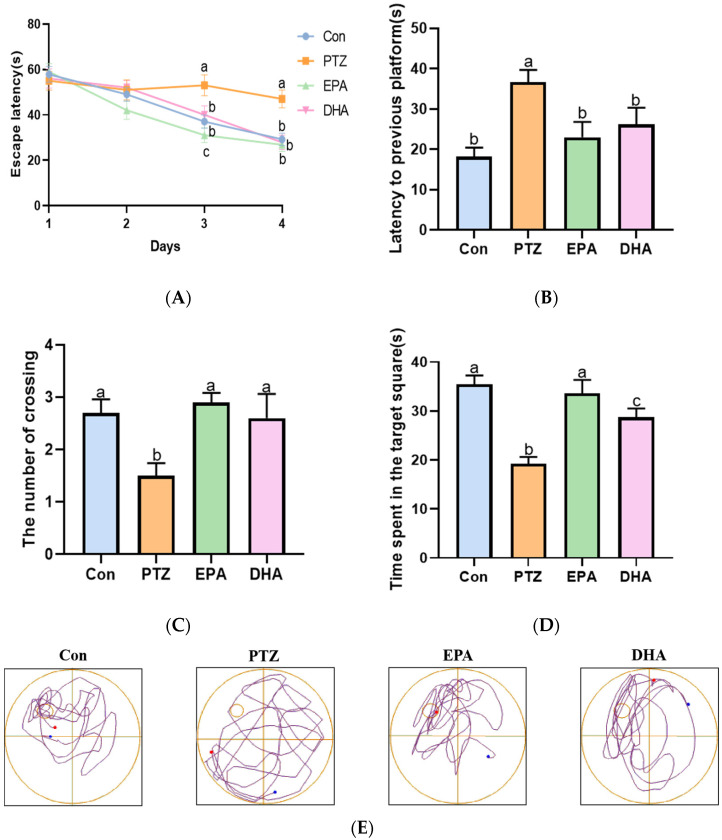
Effects of EPA and DHA on cognitive deficiency in PTZ-kindling young mice model. (**A**) The escape latency in training phase; (**B**) the latency to previous platform; (**C**) the number of platform crossings; (**D**) the time spent in the target square; (**E**) representative track plot data. The data were presented as mean + SEM (*n* = 10), and the tested groups underwent repeated measures ANOVA. The Bonferroni post hoc test was employed to calculate *p*-values based on the Bonferroni-adjusted α. There are significant differences between different letters.

**Figure 3 marinedrugs-21-00464-f003:**
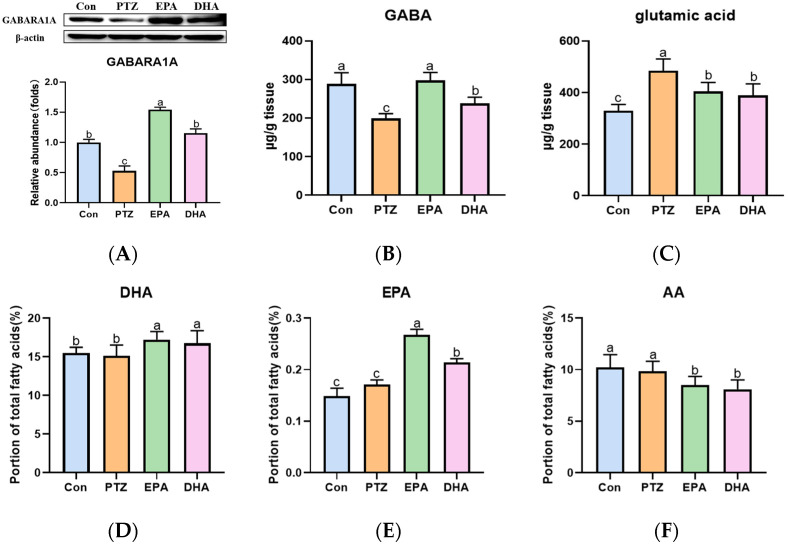
Effects of EPA and DHA on neurotransmitter disorders, as well as the levels of DHA, EPA, and arachidonic acid (AA) in the brain of young mice with PTZ kindling. (**A**) The representative Western-blots and densitometry of GABARA1 (*n* = 7). The levels of GABA (**B**), Glu (**C**) EPA (**D**), DHA (**E**), and AA (**F**) in the hippocampus (*n* = 5). Protein levels are normalized to β-actin which served as loading control and reproduced with Con group. The data were presented as mean + SEM, and the tested groups underwent repeated measures ANOVA. The Bonferroni post hoc test was employed to calculate *p*-values based on the Bonferroni-adjusted α. There are significant differences between different letters.

**Figure 4 marinedrugs-21-00464-f004:**
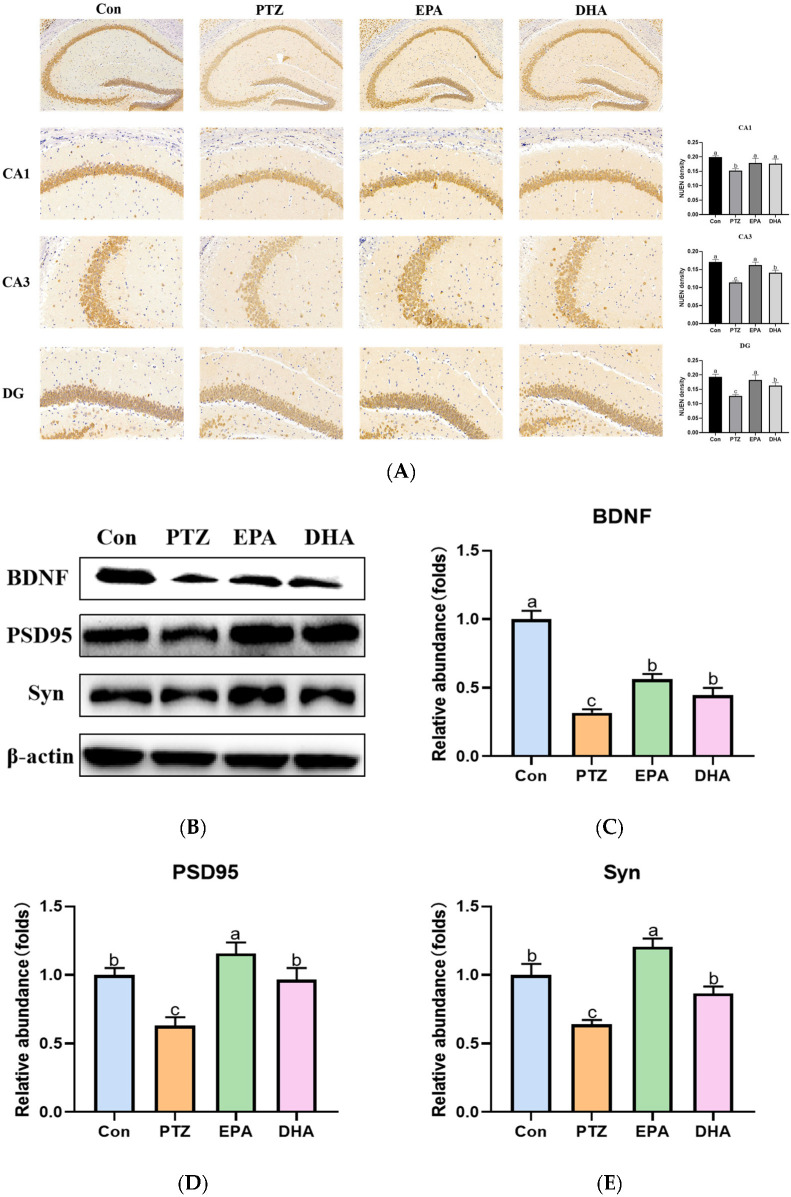
Effects of EPA and DHA on neuronal damage in PTZ-kindling young mice model. (**A**) The representative images of NeuN immunohistochemistry staining, the pixel density of NeuN staining were used to evaluate the NeuN expression among the groups. Data are presented as the mean + SEM (*n* = 3); Scale bar, 50 µm. (**B**) Representative Western-blots and (**C**–**E**) densitometry of BDNF, PSD95 and Syn. Protein levels are normalized to β-actin which served as loading control and reproduced with Con group. Values are indicated as the mean + SEM (*n* = 7), *p* < 0.05 was considered to indicate statistically significant. The tested groups underwent repeated measures ANOVA. The Bonferroni post hoc test was employed to calculate *p*-values based on the Bonferroni-adjusted α. There are significant differences between different letters.

**Figure 5 marinedrugs-21-00464-f005:**
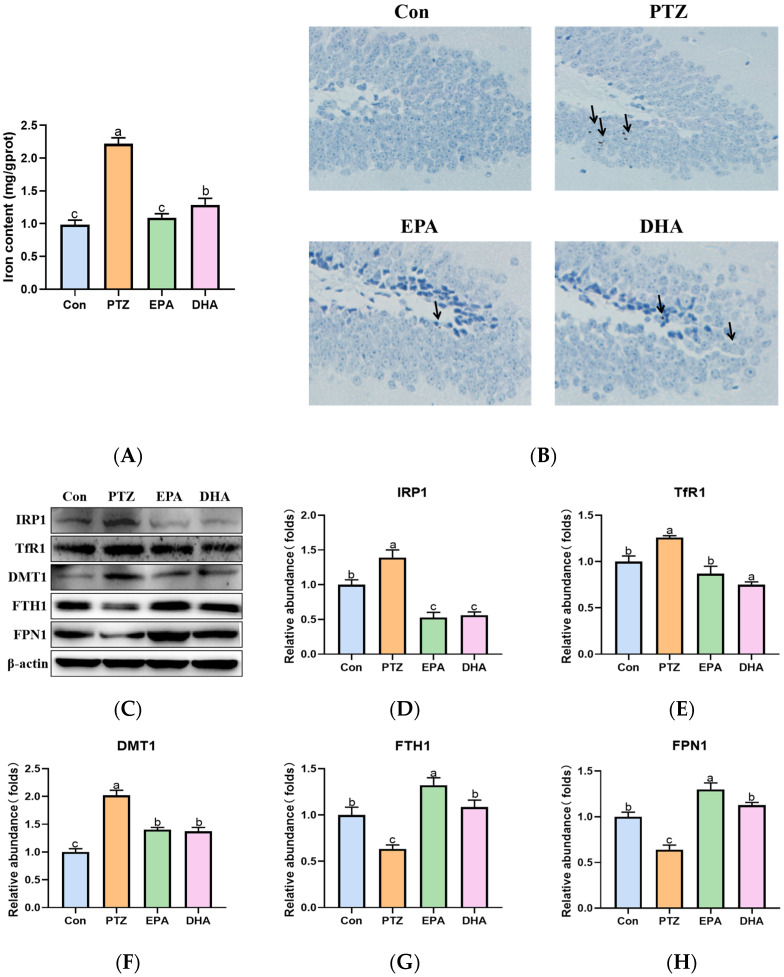
Effects of EPA and DHA on hippocampal iron dyshomeostasis in PTZ kindling young mice model. (**A**) Total iron content in the hippocampus; (**B**) DAB staining in the hippocampus (*n* = 3); Scale bar, 200 µm; (**C**) representative Western-blots and (**D**–**H**) densitometry of IRP1, TfR1, DMT1, FTH1, and FPN1. Protein levels are normalized to *β*-actin which served as loading control and reproduced with Con group. Values are indicated as the mean + SEM (*n* = 7). The tested groups underwent repeated measures ANOVA. The Bonferroni post hoc test was employed to calculate *p*-values based on the Bonferroni-adjusted α. There are significant differences between different letters.

**Figure 6 marinedrugs-21-00464-f006:**
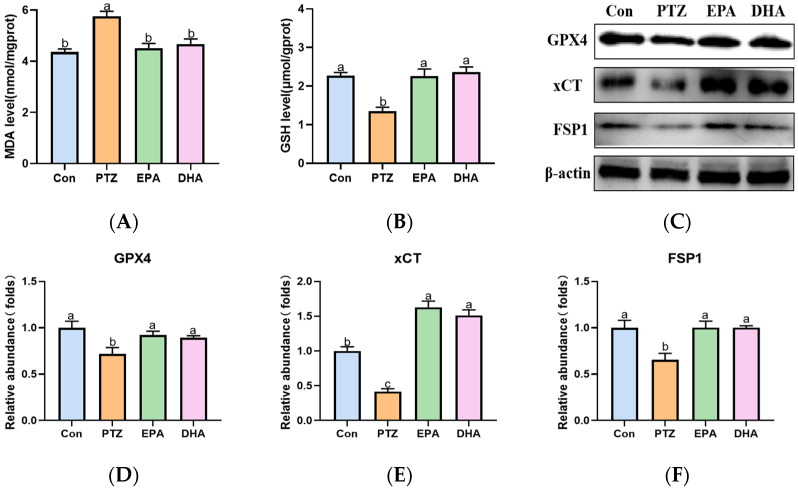
Effects of EPA and DHA on lipids peroxidation in PTZ kindling young mice model. (**A**) The level of MDA, (**B**) GSH and representative western-blots (**C**) and densitometry of GPX4, xCT and FSP1 (**D**–**F**). Protein levels are normalized to *β*-actin which served as loading control and reproduced with Con group. Values are indicated as the mean + SEM (*n* = 7). The tested groups underwent repeated measures ANOVA. The Bonferroni post hoc test was employed to calculate *p*-values based on the Bonferroni-adjusted α. There are significant differences between different letters.

**Figure 7 marinedrugs-21-00464-f007:**
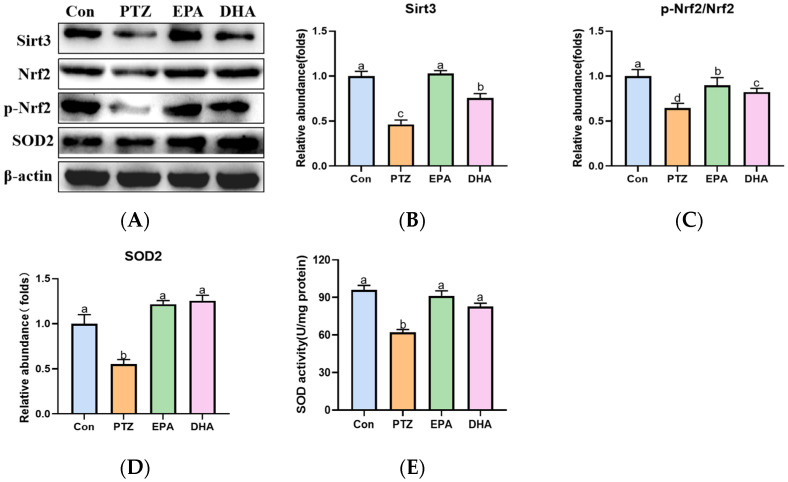
Effects of EPA and DHA on Sirt3/Nrf2 pathway in PTZ-kindling young mice model. (**A**) The activity of SOD; (**B**) representative Western-blots and (**C**–**E**) densitometry of Sirt3, *p*-Nrf2/Nrf2 and SOD2. Protein levels are normalized to *β*-actin which served as loading control and reproduced with Con group. Values are indicated as the mean + SEM (*n* = 7). The tested groups underwent repeated measures ANOVA. The Bonferroni post hoc test was employed to calculate *p*-values based on the Bonferroni-adjusted α. There are significant differences between different letters.

**Figure 8 marinedrugs-21-00464-f008:**
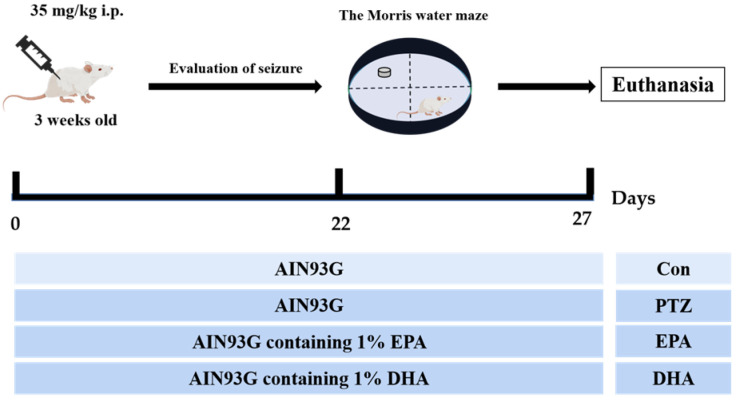
Experimental design and animal administration. Mice were intraperitoneally injected with pentetrazol (PTZ) every other day for a total of 11 times.

## Data Availability

Not applicable.

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
