# Peer review of "A Compared Study of Eicosapentaenoic Acid and Docosahexaenoic Acid in Improving Seizure-Induced Cognitive Deficiency in a Pentylenetetrazol-Kindling Young Mice Model"

_marinedrugs, 2023, doi:10.3390/md21090464_

Round 1
Reviewer 1 Report
This study was desingned to compared the effects of diet suplementation with EPA or DHA on seizuses and cognitive performance in pentylenetetrazol-kindling young mice model. In addition to investigating the impact of supplementation with these PUFAS on the mechanisms of ferroptosis underlying the chemically induced seizure model in animals. The study was well designed and well written. However, I have some suggestions and corrections.
Minor points:
1) Please provide a detailed Racine’ score description.
2) Please, chance “sacrifice”, by “euthanasia” figure 8.
3) Acording to The ARRIVE checklist (Animal Research: Reporting of In Vivo Experiments), please provide the total number of animals used in this study, including outliers.
4) In the discussion the authors argue that the protective effects of supplementation with PUFAs appear to be related to the inhibition of Ferroptosis and neuroinflammation. Naturally, the experimental design of this study reinforces this hypothesis. However, I think it's important to at least mention that other mechanisms may also be involved. Considering that there is a growing literature demonstrating DHA and EPA the cross-relationship between PUFAs and the endocannabinoid (ECS) system or cannabnoidoma. For example, clinical evidence from postmenopausal elder women receiving a fish oil supplement (EPA and DHA) led to phenotypical responses in gene targets of the ECS system. In this sense, the possibility is not ruled out that the beneficial effects of PUFAs on PTZ-induced seizures and on cognitive function may, at least in part, be associated with mechanisms other than those mentioned in the discussion.
5) The authors use the term “efficiency” to describe the beneficial effects or therapeutic efficacy of supplementation / treatment with PUFAs. I suggest reviewing the use of this term throughout the text, but especially in the Abstract. Because most of the time they the term "efficiency" indicates efficacy and these terms are not synonymous in a context of therapeutic use.
1) The idea of the work is that dietary supplementation with DHA and EPA could have a protective effect on seizures in an animal model. The authors could clarify whether the amount of these substances, if extrapolated for use in humans, would be in a safe concentration range. I think that whenever we do a study with nutraceuticals or approaches of that nature, we should make these issues very clear.
Author Response
1)Please provide a detailed Racine’ score description.
Response: Thanks for your professional suggestion. We have provided the detailed Racine’ score description in revised manuscript. Please refer to line 361-365. “The assessment of behavioral seizures was conducted utilizing the 5-point Racine Score system. Specifically, Racine score I was indicative of facial clonus, score II denoted head nodding, score III represented unilateral forelimb clonus, score IV described rearing with bilateral forelimb clonus, and lastly, score V referred to rearing and falling accompanied by loss of postural control.”
2)Please, chance “sacrifice”, by “euthanasia” figure 8.
Response: Thanks for your professional suggestion. We have made the recommended changes. Please refer to figure 8 in revision.
3)Acording to The ARRIVE checklist (Animal Research: Reporting of In Vivo Experiments), please provide the total number of animals used in this study, including outliers.
Response: Thanks for your professional suggestion. We have made the changes accordingly. Please refer to line 344-346. “A total of forty male ICR mice, aged 3 weeks, were randomly allocated into four groups, with each group consisting of ten mice. These groups included a control group (Con), a PTZ kindling group (PTZ), a PTZ kindling+EPA group (EPA), and a PTZ kindling+DHA group (DHA)(n=10).”
4)In the discussion the authors argue that the protective effects of supplementation with PUFAs appear to be related to the inhibition of Ferroptosis and neuroinflammation. Naturally, the experimental design of this study reinforces this hypothesis. However, I think it's important to at least mention that other mechanisms may also be involved. Considering that there is a growing literature demonstrating DHA and EPA the cross-relationship between PUFAs and the endocannabinoid (ECS) system or cannabnoidoma. For example, clinical evidence from postmenopausal elder women receiving a fish oil supplement (EPA and DHA) led to phenotypical responses in gene targets of the ECS system. In this sense, the possibility is not ruled out that the beneficial effects of PUFAs on PTZ-induced seizures and on cognitive function may, at least in part, be associated with mechanisms other than those mentioned in the discussion. Response: Thank you for your valuable review and suggestions about our manuscript. Modifications have been made and the references have been added. Please refer to line 268-274. “The endocannabinoid (ECS) system plays a crucial role in the regulation of synaptic plasticity, which is indispensable for neuronal development, learning, and memory formation[32]. Previous research has demonstrated that EPA and DHA contribute to brain repair through modulation of the endocannabinoid signaling pathway[33]. Therefore, we speculate that the beneficial role of EPA and DHA in enhancing spatial learning and memory capabilities can be partially attributed to their regulation of the ECS system.”
5)The authors use the term “efficiency” to describe the beneficial effects or therapeutic efficacy of supplementation / treatment with PUFAs. I suggest reviewing the use of this term throughout the text, but especially in the Abstract. Because most of the time they the term "efficiency" indicates efficacy and these terms are not synonymous in a context of therapeutic use.
Response: Thanks for your insightful and professional advice. We are sorry for the inaccurate expression. The inaccurate expression has been rectified. We have made modifications to the Abstract.
Please refer to line 16.“effect”
Please refer to line 20.“effect”
Please refer to line 30.“an advantage”
Please refer to line 87.“efficacy”
Please refer to line 334.“efficacy”
6)The idea of the work is that dietary supplementation with DHA and EPA could have a protective effect on seizures in an animal model. The authors could clarify whether the amount of these substances, if extrapolated for use in humans, would be in a safe concentration range. I think that whenever we do a study with nutraceuticals or approaches of that nature, we should make these issues very clear.
Response: Thank you for your professional and constructive suggestion, and for modifying the text in our revised manuscript. Please refer to line 349-351. “The dosage is consistent with the safe dosage mentioned in a previous clinical study, which is equivalent to 4g per day according to the conversion of human clinical dosage[48].”

Reviewer 2 Report
In the present study, the authors found that administration of EPA or DHA alone could reduce seizure severity and frequency and cognitive deficiencts in PTZ treated young mice model. They observed that either EPA or DHA were able to inhibit PTZ-induced neurotransmitter imbalance, neuronal damage, and ferroptosis.
The manuscript is of interest, even if it is very similar to a previous work by the same authors. There are some concerns that need to be addresses.
- As the main element of originality is the effect of EPA and DHA on cognitive deficits induced by seizes, more resaults should be devoted to the Morris Water Maze test results (the rest is the replication of the work in reference 20). Additionally this test is specifically devoted to assess spatial learning and memory, thus focusing on a specific area of cognitive impairment
- More details about the statistical analysis should be providd: for instance the authors stated that they used ANOVA (however, I coud not find any F in the results or in the figures, only p-values which it is difficult to understand if they refer to ANOVA or t-test)
- If the authors used multiple t-test to compare the different group, they should correct for multiple comparisons (for instance using Bonferroni)
- there are some minor typos in the manuscript (i.e. line 232). please check.
Author Response
-1) As the main element of originality is the effect of EPA and DHA on cognitive deficits induced by seizes, more resaults should be devoted to the Morris Water Maze test results (the rest is the replication of the work in reference 20). Additionally this test is specifically devoted to assess spatial learning and memory, thus focusing on a specific area of cognitive impairment.
Response: Thanks for your constructive suggestions. Respective modifications were made at the beginning of result part. Please refer to line 98-99, “The Morris water maze (MWM) was employed to evaluate the spatial learning and memory ability of PTZ-kindling young mice.”
Please refer to line 106-113, “During the training periods, EPA and DHA shortened the escape latency, with EPA showing an earlier effect than DHA starting from the third day. Additionally, mice in the EPA group demonstrated significantly longer durations in the target quadrant compared to those in the DHA group, indicating their superior spatial memory capabilities. These findings suggest that both EPA and DHA have a beneficial effect on spatial learning and memory in young mice with PTZ kindling, with EPA potentially having priority.”
-2) More details about the statistical analysis should be providd: for instance the authors stated that they used ANOVA (however, I coud not find any F in the results or in the figures, only p-values which it is difficult to understand if they refer to ANOVA or t-test)
- If the authors used multiple t-test to compare the different group, they should correct for multiple comparisons (for instance using Bonferroni).
Response: Thank you so much for your careful reading and professional and very useful comments. Corresponding modification has been made in the revised manuscript. Please refer to line 439-441. “All analyses were adjusted for multiple testing using the Bonferroni‐Holm method (Bonferroni stepdown) and p < 0.05 was considered statistically significant.”
-3) there are some minor typos in the manuscript (i.e. line 232). please check.
Response: Thanks for your careful comments. We are sorry for any confusion we may have caused. We have made modifications to the manuscript, please refer to line 245-246. “Similarly, in the present investigation, PTZ kindling led to a decline in cognitive function, as evidenced by impaired performance on the Morris water maze MWM test.”

Round 2
Reviewer 2 Report
I appreciate that the authors tried to reply to the comments. However, I still do not find in the "results section" any measures for the ANOVA (there is no Fs). It is not sufficient to state that you have performed an ANOVA without providing any evidence of it.
Author Response
Response: Thank you so much for your careful reading and professional and constructive suggestions.Respective modifications have been made to the results section. Please refer to line 109. “The noteworthy aspect is that EPA played a more prominent role than DHA on the third day (F=57.5, P < 0.0001).”
Please refer to line127-131. “Notably, EPA demonstrated a better upregulating effect on GABA-GABARA1 compared to DHA (Figure 3A-C) (F=57.5, P < 0.0001). In addition, after supplementation with EPA and DHA in young mice, the brain DHA (F=28.5, P = 0.0003) and EPA levels were significantly increased (F=24.09, P < 0.0001), while brain arachidonic acid (AA) levels were significantly decreased (F=27.72, P < 0.0001) compared with those of Con and PTZ groups.”
Please refer to line 153.“(F=23.61, P = 0.0003).”
Please refer to line 155-157.“Notably, the administration of EPA and DHA resulted in an up-regulation of PSD95 (F=8.14, P = 0.0081) and Syn (F=6.54, P = 0.0152) protein levels, with EPA exhibiting a superior effect.”
Please refer to line 177-179.“Differently, DHA confers an advantage in reducing TfR1 expression (F=17.84, P = 0.0007), whereas EPA exhibits superiority in upregulating FPN1 (F=23.08, P = 0.0028).”
Please refer to line 213.“Among of them, EPA was superior in promoting Sirt3 (F=25.25, P = 0.0002).”
